# PIKA: EXPERT-LEVEL SYNTHETIC DATASETS FOR POST-TRAINING ALIGNMENT FROM SCRATCH

## ABSTRACT

Reinforcement Learning from Human Feedback (RLHF) has become a corner-stone for aligning large language models (LLMs). However, its effectiveness critically depends on high-quality instruction data. Most existing high-quality alignment datasets are either private or require costly human annotation, which hinders reproducibility and scalability. Even with the emergence of Reinforcement Learning from AI Feedback (RLAIF), concerns about data quality remain. More-over, it is still unclear to the open-source community how much data is actually required to fine-tune a base model into a strong instruction-following model. Cur-rent state-of-the-art approaches typically rely on over 300k examples even at the supervised fine-tuning (SFT) stage, yet they still underperform compared to propri-etary models, leaving substantial barriers for academic and resource-constrained communities. To address this gap, we introduce PiKa, a data-efficient family of expert-level alignment datasets. In particular, the PiKa-SFT dataset uses only 30k SFT examples—an order of magnitude fewer than the SoTA dataset Magpie. Through extensive evaluations by fine-tuning Llama-3-8B-Base on PiKa and other public instruction following datasets, we show that PiKa-SFT alone outperforms models trained on much larger datasets. Remarkably, on two widely used alignment benchmarks, AlpacaEval 2.0 and Arena-Hard, PiKa-SFT fine-tuning surpasses the official Llama-3-8B-Instruct model – which was trained on over 10M proprietary examples. We further extend our study by training the Qwen2.5 series (0.5B–7B) on PiKa-SFT, consistently outperforming their official instruction-tuned counter-parts. In addition, we curate 30k high-quality preference optimization examples, which further improve alignment performance when applied after SFT initializa-tion. These findings demonstrate that high-quality alignment can be achieved with significantly reduced data, providing a practical and scalable path for advancing open-source LLM alignment research. Our code and data will be available at https://anonymous.4open.science/r/PiKa.

## 1 INTRODUCTION

Large language models (LLMs) have rapidly become the foundation of modern AI systems, achieving strong performance across a broad spectrum of instruction-following tasks (Achiam et al., 2023; Meta, 2024; Qwen et al., 2025). A key factor behind this success lies in instruction fine-tuning, which enables models to generalize beyond their pretraining distribution and handle novel user queries effectively. The quality of this instruction tuning, however, is highly dependent on the availability of reliable alignment datasets. Despite the release of powerful open-source models, the datasets used to align models remain largely proprietary. This lack of transparency poses a major obstacle to reproducibility and hinders progress in open research on improving LLM alignment (Xu et al., 2025).

To overcome the difficulty of building high-quality datasets, two major directions have been explored. The first relies on human experts to author and curate instruction–response pairs (Databricks, 2023; Köpf et al., 2023; Zhao et al., 2024; Zheng et al., 2024; 2023), a process that is both *time-consuming* and *labor-intensive* (Liu et al., 2024). The second line of work leverages LLMs themselves to automatically generate synthetic data (Ding et al., 2023; Yin et al., 2023; Li et al., 2024a; Sun et al., 2023; Taori et al., 2023; Wang et al., 2023; 2024; Xu et al., 2023a;b; Li et al., 2023a). More recently, efforts have shifted toward increasing the *diversity* of synthetic datasets and scaling them up to massive collections of conversational examples that aim to approximate human interaction. However, despite such progress, models trained from scratch on these large-scale synthetic corpora often

**PiKa Dataset Construction Pipeline**

Figure 1: Overview of the PIKA pipeline for synthesizing expert-level alignment data. **Step 1:** An expert-level persona covering multiple domains is given to an aligned LLM, which auto-regressively generates high-quality and knowledge-intensive instructions. **Step 2:** For each instruction, the LLM produces multiple candidate responses. **Step 3:** A reward model scores these responses. For SFT, we select the instruction and its highest-scoring response as a training pair. For DPO, we construct preference data by pairing the highest- and lowest-scoring responses with the same instruction.

acquire only limited instruction-following ability, and the training process itself requires significant computational resources, making it less accessible for much of the research community.

*Do we really need such massive datasets to enhance the instruction-following ability of base models?* Intuitively, training on large collections of diverse but overly simple data pairs provides only limited informational gain, as models tend to mimic surface-level patterns rather than generalize to more complex scenarios. Even when prompts and responses are "high-quality", prompt difficulty largely determines how much depth of knowledge the model utilizes during optimization. In other words, datasets containing more challenging instructions of comparable quality require richer reasoning and more detailed solutions, which can improve performance not only on difficult tasks but also transfer effectively to simpler ones. Thus, the utility of an instruction dataset is not merely a matter of scale, but of how well its prompts capture the complexity needed to guide robust instruction following.

Building on these insights, we construct an expert-level synthetic instruction dataset named **PiKa**, as illustrated in Figure 1. Our approach is inspired by persona-based generation methods (Ge et al., 2025), which prompt LLMs to role-play as specific personas to generate diverse instructions. To further enhance data quality, we rigorously sample and filter persona prompts, removing harmful or policy-violating cases and retaining only complex personas for instruction generation. Unlike many existing datasets, PiKa is fully synthesized using GPT-4o, a model widely regarded as both highly capable and human-like, ensuring consistently high-quality outputs. Moreover, all generated instruction–response pairs are filtered and re-scored by GPT-4o, resulting in datasets dominated by challenging, expert-level prompts with detailed solutions. We provide comprehensive statistics, highlight the advantages of PiKa, and present an in-depth analysis in Section 3, enabling practitioners to flexibly select and filter data for fine-tuning according to their specific needs.

To evaluate the effectiveness of PiKa, we compare it with widely used public instruction datasets (e.g., ShareGPT (Chiang et al., 2023), WildChat (Zhao et al., 2024), UltraChat (Ding et al., 2023), OpenHermes (Teknium, 2023a;b), Tulu V2 Mix (Ivison et al., 2023), and Magpie-Air/Pro (Xu et al., 2025)) by conducting supervised fine-tuning (SFT) of the Llama-3-8B-Base model on each dataset and evaluating the resulting models on standard alignment benchmarks such as AlpacaEval 2 (Li et al., 2023b) and Arena-Hard (Li et al., 2024b), following recent Magpie studies. Our results show that models fine-tuned on PiKa consistently achieve superior performance, even surpassing those trained with both SFT and direct preference optimization (DPO) (Rafailov et al., 2023) on UltraFeedback (Cui et al., 2023). Remarkably, PiKa is more than an order of magnitude smaller than Magpie-Pro (30K vs. 300K examples for SFT). Despite this significant difference in dataset size, PiKa achieves superior performance across benchmarks. For the first time, our approach enables open-source aligned models to significantly outperform the official Llama-3-8B-Instruct on both AlpacaEval 2.0

and Arena-Hard. This is particularly remarkable given that Llama-3-8B-Instruct was trained with over 10M proprietary examples and subsequent preference optimization. Building on this, we extend our study to the Qwen2.5 (0.5B–7B), where models trained on PiKa-SFT consistently surpass their official instruction-tuned counterparts.

Furthermore, PiKa not only excels under SFT alone compared to prior public datasets, but also delivers state-of-the-art results when combined with preference optimization methods such as DPO. By extending PiKa to generate high-quality preference optimization data, PiKa-aligned Llama-3 models achieve substantial improvements and outperform models trained with both the Magpie and UltraFeedback series, with particularly strong gains on the more challenging Arena-Hard benchmark (43.70 vs. 33.30 for Magpie-Pro and 25.30 for UltraFeedback respectively). These findings highlight the exceptional quality and efficiency of PiKa, demonstrating that carefully constructed synthetic data can outperform extensively optimized proprietary datasets in a far more data-efficient manner.

## 2 PIKA: EXPERT-LEVEL SYNTHETIC DATASETS

**Step 1: Expert-Level Instruction Generation.** We sample a diverse set of complex personas $\{\pi_i\}_{i=1}^N$ from PersonaHub (Ge et al., 2025), covering multiple domains such as biology, engineering, medicine, and law. Each persona $\pi_i$ is then provided to GPT-4o, which auto-regressively generates a corresponding knowledge-intensive instruction. Based on our preliminary experiments, we find that more domain-specific and sophisticated personas tend to generate more challenging prompts:

$$I_i = \text{LLM}(\pi_i), \quad i = 1, \dots, N. \tag{1}$$

To ensure diversity, each persona is used only once. Generated instructions are further filtered through a quality control sampling $Q(I_i) \in \{0, 1\}$, retaining only those with $Q(I_i) = 1$, i.e., instructions that are challenging, safe, and informative.

**Step 2: Multi-Path Response Generation.** For each validated instruction $I_i$, the LLM produces $k$ candidate responses under mild stochasticity with temperature $T < 1$:

$$R_i = \{r_{i1}, r_{i2}, \dots, r_{ik}\} = \text{LLM}(I_i; T). \tag{2}$$

This yields diverse data pairs $(I_i, r_{ij})$ that differ in reasoning depth, style, and completeness.

**Step 3: Reward-Model-Guided Selection.** For each instruction $I_i$ ($i = 1, \dots, N$), a reward model $\mathcal{R}(\cdot)$ assigns a score to each candidate responses. In our default setting, we employ `Skywork-Reward-V2-Llama-3.1-8B` (Liu et al., 2025), which ranks first on the Reward-Bench leaderboard[1]. The reward score for each candidate response $r_{ij}$ of instruction $I_i$ is computed as:

$$s_{ij} = \mathcal{R}(I_i, r_{ij}), \quad j = 1, \dots, k, \tag{3}$$

where $k$ is the number of candidate responses per instruction.

For supervised fine-tuning (SFT), we retain the pair $(I_i, r_{ij^*})$ with

$$j^* = \arg\max_j s_{ij}. \tag{4}$$

For preference optimization (e.g., DPO), we construct triples $(I_i, r_{ij^+}, r_{ij^-})$ by selecting the highest- and lowest-scoring responses:

$$j^+ = \arg\max_j s_{ij}, \quad j^- = \arg\min_j s_{ij}. \tag{5}$$

These preference pairs provide supervision for optimizing models toward expert-level alignment.

## 3 DATASET ANALYSIS

We conduct a comprehensive analysis of the proposed PiKa dataset, covering instruction similarity, length statistics, difficulty, quality, feasibility, and construction cost. This study demonstrates that PiKa delivers more challenging, diverse, and practically useful supervision compared to existing datasets such as Magpie-Pro.

---

[1] `https://huggingface.co/spaces/allenai/reward-bench`, as of 2025/9/25

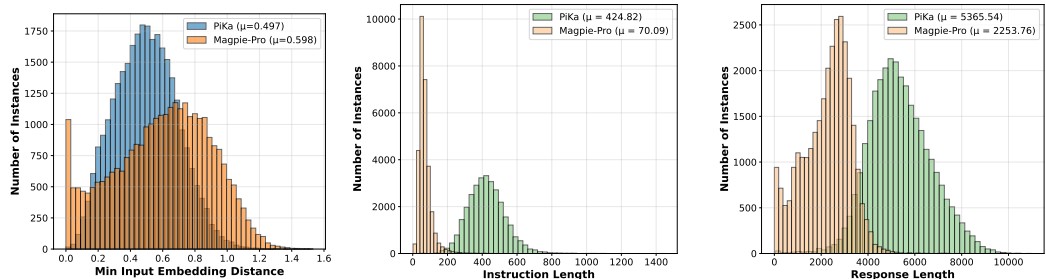

Figure 2: Comparison between PiKa and Magpie-Pro. (a) Minimum input embedding distances distribution. (b) Instruction lengths distribution. (c) Response lengths Distribution.

### 3.1 INSTRUCTION SIMILARITY

To evaluate dataset coverage, we follow prior work (Zhao et al., 2024; Xu et al., 2025) and compute the *minimum neighbor distance* (MND) in the embedding space. Instructions are encoded with `all-mpnet-base-v2`, and pairwise distances are calculated with FAISS (Douze et al., 2024). The MND for an instruction $I_i$ is defined as:

$$\text{MND}(I_i) = \min_{j \neq i} \|e(I_i) - e(I_j)\|_2, \tag{6}$$

where $e(I)$ denotes the embedding of instruction $I$. Larger MND indicates lower redundancy and higher diversity. As shown in Figure 2(a), PiKa achieves a mean MND of 0.497 compared to Magpie-Pro's 0.598, indicating comparable diversity while effectively avoiding repetitive prompts that plague many existing datasets.

### 3.2 LENGTH ANALYSIS

We further analyze tokenized lengths of instructions and responses. Figure 2(b, c) shows that PiKa instructions average around 424 tokens, and responses exceed 5,305 characters on average—substantially longer than existing datasets—indicating that PiKa provides more detailed and knowledge-intensive supervision.

### 3.3 DATASET ASSESSMENT

Beyond coverage and length, we adopt GPT-4o as an expert evaluator to assess *difficulty*, *feasibility*, and *quality*. Each attribute is scored on a 1 to 10 scale with dedicated prompts (see Appendix C). Formally, let

$$s_d : \mathcal{X} \to [1, 10], \tag{7}$$
$$s_f : \mathcal{X} \to [1, 10], \tag{8}$$
$$s_q : \mathcal{X} \times \mathcal{Y} \to [1, 10], \tag{9}$$

denote the difficulty of instruction $x$, the feasibility of instruction $x$, and the quality of instruction response pair $(x, y)$, respectively.

#### 3.3.1 DIFFICULTY EVALUATION

We evaluate instruction difficulty based on task complexity, required domain knowledge, cognitive load, and technical sophistication. The framework categorizes tasks into elementary knowledge (scores 1–2), specialized expertise (3–6), expert-level understanding (7–8), and cutting-edge research knowledge (9–10). As shown in Figure 3(a), PiKa instructions exhibit significantly higher difficulty, with a mean score of 7.39 compared to Magpie-Pro's 2.65. The distribution further reveals that while Magpie-Pro concentrates heavily in the elementary to intermediate range (scores 1–4), PiKa covers a broader spectrum, with a substantial proportion requiring expert-level reasoning and advanced

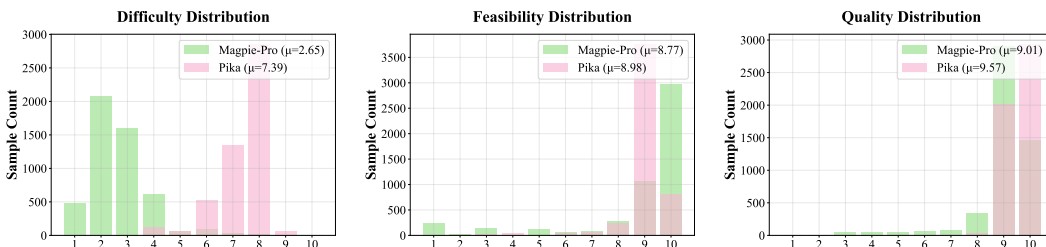

Figure 3: GPT-4o–based evaluation of PiKa's instruction difficulty, feasibility and instruction-response quality compared with Magpie-Pro.

**Magpie-Pro**

**Instruction:** *"Write a poem about the beauty of the ocean."*

**Difficulty Score: 2**

**PiKa**

**Instruction:** *"Analyze the impact of international sanctions and oil price fluctuations on the Russian economy in 2015. Discuss how these factors influenced inflation, recession risks, and consumer prices, particularly in relation to the agricultural and food import ban. Additionally, evaluate the potential for massive withdrawals from the banking system and their broader economic consequences."*

**Difficulty Score: 7**

Figure 4: Example instructions from Magpie-Pro and PiKa datasets. PiKa instructions are generally more complex and domain-specific, while Magpie-Pro tends to focus on simpler, open-ended prompts. We also include detailed instruction-response pairs in Appendix D.

technical understanding. This demonstrates that PiKa effectively captures more sophisticated tasks, and Figure 4 provides illustrative comparisons.

### 3.3.2 FEASIBILITY EVALUATION

Feasibility assessment examines whether instructions represent realistic, executable tasks with practical applicability. The evaluation considers task feasibility, contextual reasonableness, logical consistency, and real world relevance. Scores range from completely unrealistic tasks (1 to 2) to highly practical and useful instructions (9 to 10). As demonstrated in Figure 3(b), PiKa maintains strong feasibility with a mean score of 8.98 compared to Magpie-Pro's 8.77. Notably, both datasets show high concentration in the upper feasibility ranges (scores 8 to 10), but PiKa demonstrates a slightly more consistent distribution in the high feasibility region. This ensures that despite the increased difficulty, PiKa instructions remain grounded in practical, real world applications that users would genuinely encounter.

### 3.3.3 QUALITY EVALUATION

Response quality is evaluated based on accuracy, completeness, clarity, instruction response alignment, educational value, and real world applicability. The scoring ranges from poor quality with incorrect information (1 to 2) to excellent quality with comprehensive and insightful responses (9 to 10). Figure 3(c) shows that PiKa achieves superior response quality with a mean score of 9.57 compared to Magpie-Pro's 9.01. Both datasets demonstrate high quality responses concentrated in the 8 to 10 range, but PiKa shows a more pronounced peak at score 10, indicating a higher proportion of exceptional responses. The quality improvement, combined with significantly longer response lengths, suggests

that PiKa responses not only meet accuracy standards but also provide more thorough, comprehensive, and educationally valuable content for alignment training.

Table 1: Performance comparison of instruction-tuned models based on Llama-3-8B using PiKa-generated versus baseline datasets. PiKa achieves superior performance while requiring $10\times$ less training data than state-of-the-art Magpie methods. Standard deviations (SD) reported are computed across both LC and WR metrics and #Convs indicates the number examples used for training the base model.

| Alignment Setup (Base LLM = **Llama-3-8B**) | #Convs | AlpacaEval 2 | | | | | | Arena-Hard GPT-4 (0314) |
| | | GPT-4-Turbo (1106) | | | Llama-3-8B-Instruct | | | |
| | | LC (%) | WR (%) | SD | LC (%) | WR (%) | SD | WR(%) |
| Llama-3-8B-Instruct (SFT+DPO) | >1M | 28.36 | 27.93 | 1.48 | 50.0 | 50.0 | - | 24.5 |
| Self-Instruct (Llama-3) (Wang et al., 2023) | 100K | 8.86 | 4.16 | 0.66 | 24.48 | 11.97 | 1.09 | 3.3 |
| ShareGPT (Chiang et al., 2023) | 112K | 6.98 | 4.00 | 0.64 | 26.05 | 15.98 | 1.22 | 6.9 |
| Ultrachat (Ding et al., 2023) | 208k | 6.70 | 3.48 | 0.60 | 24.14 | 13.37 | 1.14 | 3.6 |
| OpenHermes 1 (Teknium, 2023a) | 243K | 8.69 | 4.98 | 0.70 | 26.81 | 16.98 | 1.25 | 5.3 |
| Tulu V2 Mix (Ivison et al., 2023) | 326k | 10.95 | 6.43 | 0.81 | 24.84 | 15.39 | 1.20 | 6.3 |
| WildChat (Zhao et al., 2024) | 652K | 14.75 | 8.43 | 0.91 | 33.99 | 23.77 | 1.42 | 11.7 |
| OpenHermes 2.5 (Teknium, 2023b) | 1M | 12.40 | 7.80 | 0.87 | 37.14 | 27.86 | 1.48 | 7.7 |
| MAGPIE-Air-300K-Filtered (Xu et al., 2025) | 300K | 25.24 | 27.33 | 1.47 | 43.79 | 47.35 | 1.69 | 20.7 |
| MAGPIE-Pro-300K-Filtered (Xu et al., 2025) | 300K | 24.06 | 28.60 | 1.51 | 41.28 | 46.80 | 1.68 | 23.9 |
| **PIKA (Ours)** | **30K** | **32.82** | **30.56** | **1.54** | **52.42** | **50.30** | **1.68** | **33.5** |

# 4 PERFORMANCE ANALYSIS

## 4.1 EXPERIMENTAL SETUP

**Baselines for Supervised Fine-Tuning and Preference Optimization.** We compare PiKa with nine state-of-the-art open-source instruction datasets: **Self-Instruct** (Wang et al., 2023), **ShareGPT** (Chiang et al., 2023), **WildChat** (Zhao et al., 2024), **UltraChat** (Ding et al., 2023), **OpenHermes 1** (Teknium, 2023a), **OpenHermes 2.5** (Teknium, 2023b), **Tulu V2 Mix** (Ivison et al., 2023), **Magpie-Air** (Xu et al., 2025), and **Magpie-Pro** (Xu et al., 2025). ShareGPT and WildChat represent human-authored datasets containing 112K and 652K high-quality multi-turn conversations between humans and ChatGPT, respectively. UltraChat and the Magpie family are representative open-source synthetic datasets. For preference optimization, we compare models aligned using PiKa with direct preference optimization (DPO) (Rafailov et al., 2023) baselines, specifically comparing against two state-of-the-art open-source training datasets: UltraFeedback and Magpie-Pro.

**Model Alignment Details.** We conduct experiments on the Llama-3 and Qwen2.5 base models. For supervised fine-tuning, we follow Touvron et al. (2023) and employ a cosine learning rate schedule with an initial learning rate of $2 \times 10^{-5}$. The maximum sequence length is set to 8192 tokens. For DPO training, we use a cosine learning rate of $5 \times 10^{-7}$. We adhere to the official instruction templates of each respective model architecture.

**Evaluation Benchmarks.** We evaluate aligned model performance using two widely adopted instruction-following benchmarks: AlpacaEval 2 (Li et al., 2023b) and Arena-Hard (Li et al., 2024b). AlpacaEval 2 comprises 805 representative instructions selected from real user interactions, providing a comprehensive assessment of practical instruction-following capabilities. Arena-Hard represents an enhanced version of MT-Bench (Zheng et al., 2023), containing 500 challenging user queries designed to test advanced reasoning and problem-solving abilities. Both benchmarks employ GPT-based evaluators to assess responses generated by the model under evaluation against established baseline models. Specifically, we use GPT-4-Turbo (1106) and Llama-3-8B-Instruct as baselines for AlpacaEval 2 and Arena-Hard uses GPT-4 (0314) as its baseline. In addition, we employ the latest GPT-4o as the LLM-as-judge to provide more contemporary and robust evaluation standards.

**Metrics.** We adopt two complementary metrics to measure instruction-following capabilities of fine-tuned models. The first metric is the **win rate (WR)**, which calculates the fraction of responses favored by the GPT evaluator over the baseline model. This metric is applied across both AlpacaEval 2 and Arena-Hard benchmarks. The second metric is the **length-controlled win rate (LC)** (Dubois et al., 2024), a debiased variant of WR that accounts for response length differences. The GPT

Table 2: Performance comparison of models instruction-tuned on Qwen2.5 base models using the PiKa 30K dataset versus official instruction-tuned models. PiKa consistently outperforms official models across all model sizes.

| Alignment Setup | | AlpacaEval 2 | | | | | |
| | | GPT-4-Turbo (1106) | | | Official Aligned Model as Ref. | | |
| | | LC (%) | WR (%) | SD | LC (%) | WR (%) | SD |
|---|---|---|---|---|---|---|---|
| Qwen2.5-0.5B | Qwen2.5-0.5B-Instruct | 1.30 | 0.93 | 0.32 | 50 | 50 | - |
| | Base Model + PiKa | **1.52** | **2.15** | 0.49 | **55.29** | **55.65** | 1.66 |
| Qwen2.5-1.5B | Qwen2.5-1.5B-Instruct | 3.77 | 2.21 | 0.48 | 50 | 50 | - |
| | Base Model + PiKa | **9.64** | **11.07** | 1.04 | **70.68** | **74.35** | 1.47 |
| Qwen2.5-3B | Qwen2.5-3B-Instruct | 6.82 | 5.16 | 0.72 | 50 | 50 | - |
| | Base Model + PiKa | **14.78** | **15.83** | 1.23 | **63.25** | **66.97** | 1.56 |
| Qwen2.5-7B | Qwen2.5-7B-Instruct | 29.03 | 25.99 | 1.47 | 50 | 50 | - |
| | Base Model + PiKa | **32.55** | **33.24** | 1.59 | **58.87** | **60.23** | 1.65 |

evaluator considers the lengths of responses generated by both the baseline model and the model under evaluation when computing LC, thereby reducing the potential bias introduced by response length variations. This metric is specifically applied to the AlpacaEval 2 benchmark (Li et al., 2023b).

## 4.2 EXPERIMENTAL RESULTS

**PiKa datasets significantly outperform baselines with SFT only.** Table 1 demonstrates the superior performance of Llama-3 models fine-tuned with PiKa-generated instruction datasets compared to those trained on baseline datasets. Using only 30K conversations, PiKa achieves remarkable results on AlpacaEval 2: 32.82% LC and 30.56% WR against GPT-4-Turbo, substantially outperforming all baseline SFT datasets. Most notably, PiKa surpasses the closest competitor Magpie-Pro (24.06% LC, 28.60% WR) while using 10× less data. Similar superiority is observed on Arena-Hard, where PiKa achieves 33.5% WR compared to Magpie-Pro's 23.9%.

**PiKa surpasses official aligned models with significantly fewer data.** When compared against the official Llama-3-8B-Instruct model (which underwent extensive SFT and DPO training on over 10M conversations), PiKa demonstrates exceptional efficiency. Our model achieves 52.42% LC and 50.30% WR when using Llama-3-8B-Instruct as the reference, indicating a clear preference over the officially aligned version. This represents a significant improvement over the 50% baseline, demonstrating the exceptional quality of PiKa-generated data. The Arena-Hard results further corroborate this finding, with PiKa achieving 33.5% WR compared to the official model's 24.5%.

**PiKa enhances performance across different backbone models.** Table 2 illustrates PiKa's effectiveness across the Qwen2.5 model family (0.5B to 7B), demonstrating consistent and substantial improvements across all model scales. PiKa consistently outperforms the official instruction-tuned models that have undergone both supervised fine-tuning and preference optimization, with particularly notable gains for smaller models. For Qwen2.5-0.5B, PiKa increases LC from 1.30% to 1.52% and WR from 0.93% to 2.15% against GPT-4-Turbo, representing meaningful improvements for the smallest model. The gains become more pronounced for Qwen2.5-1.5B, where PiKa achieves 9.64% LC and 11.07% WR compared to the official model's 3.77% and 2.21%, representing 2.56× and 5.01× improvements respectively. For Qwen2.5-3B, PiKa maintains strong performance with 14.78% LC versus the official 6.82% and 15.83% WR versus 5.16%. Even for the largest Qwen2.5-7B model, PiKa achieves 32.55% LC and 33.24% WR against GPT-4-Turbo, compared to 29.03% and 25.99% for the official model. When using official models as references, PiKa shows substantial improvements across all scales, with win rates ranging from 55.65% (0.5B) to 74.35% (1.5B), demonstrating that PiKa's high-quality instruction data provides consistent value regardless of the underlying model capacity, with the most dramatic relative improvements observed for smaller models.

**Performance on Preference Optimization.** To evaluate PiKa's effectiveness in preference optimization, we compare it against UltraFeedback and Magpie-Pro across data efficiency and benchmark performance (Figure 5). As shown in Figure 5(a), PiKa is highly data-efficient, requiring only 30K examples for SFT and 30K for DPO. In contrast, UltraFeedback uses 208K and 64K, while Magpie-Pro uses 300K and 60K, respectively. Despite this substantial reduction in training data, PiKa

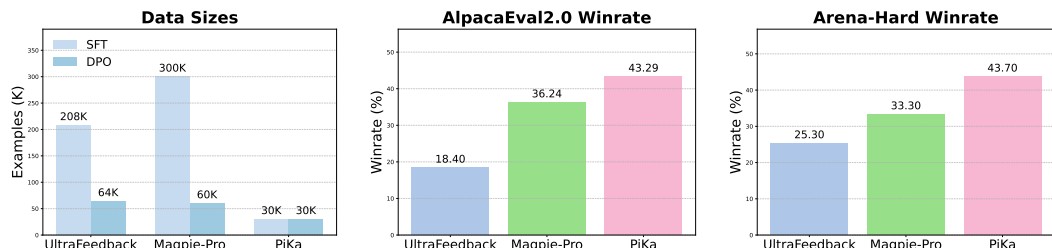

Figure 5: Comparison across three dataset families: UltraFeedback, Magpie-Pro, and PiKa. (a) Dataset sizes. (b) AlpacaEval 2.0 win rates. (c) Arena-Hard win rates. .

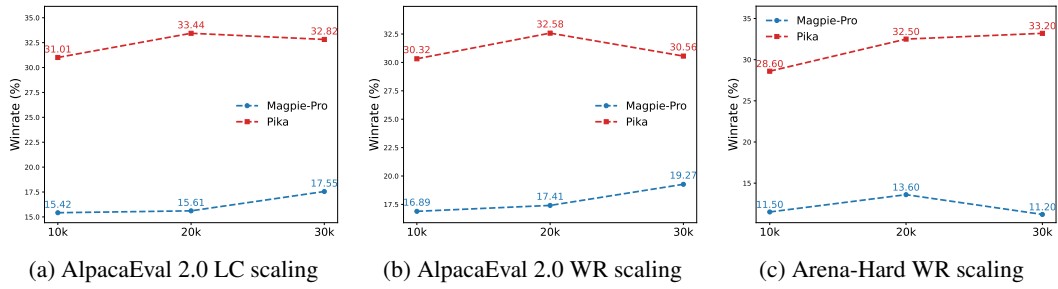

(a) AlpacaEval 2.0 LC scaling    (b) AlpacaEval 2.0 WR scaling    (c) Arena-Hard WR scaling

Figure 6: Scaling analysis of dataset size from 10K to 30K comparing PiKa versus Magpie-Pro. PiKa demonstrates consistent superiority across all data scales and maintains better performance curves, with optimal performance achieved at 30K examples on Arena-Hard.

achieves superior results on both evaluation benchmarks. On AlpacaEval 2.0, PiKa attains a 41.29% win rate, significantly surpassing Magpie-Pro (36.24%) and UltraFeedback (18.40%). Similarly, on Arena-Hard, PiKa achieves 43.70%, outperforming Magpie-Pro (33.30%) and UltraFeedback (25.30%). These results highlight PiKa's ability to generate high-quality preference data, enabling more effective alignment with human preferences even under constrained data budgets.

**Scaling Analysis.** Figure 6 presents a scaling analysis comparing PiKa and Magpie-Pro across dataset sizes from 10K to 30K. Across all metrics and scales, PiKa consistently outperforms Magpie-Pro. Notably, PiKa with only 10K examples achieves performance surpassing, Magpie-Pro with 30K examples. The scaling curves further show that PiKa reaches its peak performance at 30K examples on Arena-Hard. We therefore adopt the 30K dataset as the default setting due to its highest Arena-Hard score. Nevertheless, future work could focus more on data efficiency, as PiKa with 10K examples already demonstrates competitive performance.

**Performance on Additional Benchmarks.** Table 3 reports performance across diverse tasks from the Hugging Face Open LLM Leaderboard (Beeching et al., 2023), including MMLU (Hendrycks et al., 2020), ARC Challenge (Clark et al., 2018), HellaSwag (Zellers et al., 2019), TruthfulQA (Lin et al., 2021), WinoGrande (Levesque et al., 2012), and GSM8K (Cobbe et al., 2021). PiKa achieves competitive performance (63.53% average) comparable to other datasets, demonstrating that the quality improvements in instruction-following do not come at the expense of general capabilities. The slight performance difference on mathematical reasoning tasks (eg., GSM8K) can be attributed to the limited proportion of mathematical instructions in PiKa's current composition. Future iterations can incorporate domain-specific mathematical and coding datasets to enhance performance on reasoning-intensive tasks while maintaining PiKa's strengths in general instruction-following.

# 5  RELATED WORK

**LLM Alignment.** A central goal in large language model (LLM) research is aligning model behavior with human values and intentions. Two major approaches are instruction tuning and preference tuning. Instruction tuning (Wei et al., 2022) fine-tunes LLMs on datasets consisting of user instructions and target responses, which can be single- or multi-turn. The effectiveness of this method is closely

Table 3: Performance comparison on additional benchmarks from the Open LLM Leaderboard. All models are supervised fine-tuned on Llama-3-8B base models. Numbers in parentheses indicate the number of few-shot examples used for each task.

| Alignment Setup | MMLU (5) | ARC (25) | HellaSwag (10) | TruthfulQA (0) | WinoGrande (5) | GSM8K (5) | Average |
|---|---|---|---|---|---|---|---|
| Llama-3-8B-Instruct | 67.82 | 61.52 | 78.67 | 52.47 | 72.14 | 71.72 | **67.39** |
| ShareGPT | 66.03 | 58.45 | 81.50 | 52.34 | 74.03 | 48.67 | 63.50 |
| OpenHermes 1 | 65.42 | 62.29 | 82.15 | 50.85 | 75.61 | 47.16 | 63.58 |
| OpenHermes 2.5 | 65.70 | 61.86 | 82.53 | 51.35 | 76.09 | 67.02 | 67.09 |
| Tulu V2 Mix | 66.34 | 59.22 | 82.80 | 47.99 | 76.16 | 58.07 | 65.10 |
| WildChat | 65.95 | 59.22 | 81.39 | 53.18 | 75.30 | 48.75 | 63.97 |
| UltraChat | 65.23 | 62.12 | 81.68 | 52.76 | 75.53 | 50.57 | 64.65 |
| Magpie-Air-300K-Filtered | 64.45 | 61.01 | 79.90 | 53.48 | 72.38 | 52.24 | 63.58 |
| Magpie-Pro-300K-Filtered | 64.25 | 60.41 | 80.52 | 52.46 | 73.32 | 47.92 | 63.15 |
| **PiKa** | 62.85 | 59.98 | 80.02 | 52.48 | 73.01 | 52.84 | 63.53 |

tied to the quality and diversity of the instruction data (Taori et al., 2023; Wang et al., 2023; Zhou et al., 2023). Preference tuning, on the other hand, builds upon instruction tuning by further refining model outputs using either reinforcement learning from human feedback (RLHF) (Bai et al., 2022) or direct preference optimization methods (Azar et al., 2024; Ethayarajh et al., 2024; Hong et al., 2024; Rafailov et al., 2023). These approaches leverage datasets containing preference comparisons, enabling models to better capture nuanced human judgments.

**Persona Roleplay.** Persona-based roleplay offers a powerful mechanism for exploring the knowledge and perspectives embedded in LLMs. In this paradigm, the model's world knowledge is effectively "compressed" into distributed representations, which can then be "decompressed" through the adoption of diverse personas to generate contextually grounded text (Delétang et al., 2024; Ge et al., 2024). Recent work by Ge et al. (2025) introduced the *Persona Hub*, an automatically constructed repository derived from large-scale web data. This resource allows researchers to tap into a wide variety of perspectives within LLMs, facilitating the large-scale creation of synthetic data without reliance on a small fixed seed corpus. Compared to traditional prompt-based synthesis, persona-driven approaches can be seamlessly integrated into arbitrary prompts, leveraging the strong role-playing capacity of LLMs to generate more diverse and versatile data. In our work, persona signals are randomly sampled from the Persona Hub to construct expert-level prompts.

**Dataset Construction.** Methods for building alignment datasets generally fall into two categories: human-in-the-loop collection and synthetic instruction generation. The first line of work gathers datasets through **human interactions** with LLMs (Databricks, 2023; Zhao et al., 2024; Zheng et al., 2024; 2023; Köpf et al., 2023). While such data is often high quality, it is expensive to scale, time-consuming to curate, and may contain toxic or biased content (Zhao et al., 2024). The second line of work leverages LLMs to generate **synthetic** instructions from a small set of human-annotated seeds, which are expanded using prompting strategies (Wang et al., 2023; Taori et al., 2023; Xu et al., 2023a;b; Wang et al., 2024; Sun et al., 2023). However, these methods often suffer from limited diversity, as generated instructions tend to resemble the original seeds (Li et al., 2024a). To address this, some approaches attempt to broaden coverage by leveraging summaries of world knowledge to drive generation (Ding et al., 2023; Li et al., 2024a). Our proposed PiKa dataset belongs to the synthetic category, but differs in that we employ persona-driven prompts sampled from expert-level profiles and selectively retain high-difficulty instances to better capture preference signals.

# 6 CONCLUSION

We present PiKa, a family of expert-level alignment datasets for LLMs. By leveraging persona-driven instruction generation and automatic filtering, we curated a compact yet diverse set of SFT and preference data that improves alignment quality. Fine-tuning experiments on Llama-3-8B show that models trained on PiKa-SFT outperform those trained on much larger open-source datasets and even surpass the official Llama-3-8B-Instruct model, trained on over 10M proprietary examples. We further validate PiKa's effectiveness across different model families, such as Qwen2.5, consistently outperforming their official instruction-tuned counterparts. These results highlight PiKa's potential to make alignment research more data-efficient and accessible to the open-source community, while suggesting future directions for studying data efficiency and mixture strategies in LLM alignment.

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

## A   LIMITATION

The current version of PiKa does not include math or code reasoning data. We plan to iterate on this framework to incorporate such data in future versions. Additionally, the PiKa dataset has significant potential for size reduction. Future work can focus on more sophisticated data selection strategies to further reduce the training dataset, as we observe that 10k PiKa examples can achieve comparable performance to 30k examples during training.

## B   ETHICS STATEMENT

We carefully synthesize data using GPT-4o and filter out unsafe, harmful, or sensitive data pairs during the initial generation process. To the best of our knowledge, the resulting dataset does not contain any content that could cause harm or violate ethical standards.

## C   EVALUATION PROMPT

### C.1   DIFFICULTY EVALUATION PROMPT

We employ the following system prompt to evaluate the difficulty level of instructions:

> **System Prompt:** You are an expert evaluator for assessing the difficulty level of instructions/questions. Please evaluate the given instruction on a scale of 1-10 based on:
>
> - Complexity of the task or question
> - Level of domain knowledge required
> - Cognitive load needed to understand and process
> - Technical depth and sophistication
>
> Scoring guidelines:
>
> - 1-2: Very basic, elementary level knowledge
> - 3-4: Intermediate, requires some specialized knowledge
> - 5-6: Advanced, requires solid domain expertise
> - 7-8: Expert level, requires deep specialized knowledge
> - 9-10: Cutting-edge, requires highly specialized or research-level expertise
>
> Please respond with ONLY a single number (1-10) representing the difficulty score.

### C.2   REALISM EVALUATION PROMPT

We employ the following system prompt to evaluate the realism and feasibility of instructions:

> **System Prompt:** You are an expert evaluator for assessing the realism and feasibility of instructions/prompts. Please evaluate the given instruction on a scale of 1-10 based on:
>
> - Feasibility: Can this task actually be completed in the real world?
> - Practicality: Does this request make sense in real-world scenarios?
> - Realistic context: Are the assumptions and requirements reasonable?
> - Real-world applicability: Would someone actually need this in practice?
> - Logical consistency: Are there any contradictions or impossible requirements?
>
> Scoring guidelines:
>
> - 1-2: Completely unrealistic, impossible to execute, or nonsensical
> - 3-4: Mostly unrealistic, major feasibility issues or impractical requirements

- 5-6: Somewhat realistic but has questionable assumptions or limited applicability
- 7-8: Quite realistic and feasible, minor issues but generally practical
- 9-10: Highly realistic, completely feasible, and practically useful

Please respond with ONLY a single number (1-10) representing the realism score.

## C.3 QUALITY EVALUATION PROMPT

We employ the following system prompt to evaluate the quality of instruction-response pairs:

**System Prompt:** You are an expert evaluator for assessing the quality of instruction-response pairs. Please evaluate the given instruction-response pair on a scale of 1-10 based on:

- Accuracy and correctness of the response
- Completeness and thoroughness of the answer
- Clarity and coherence of explanation
- Alignment between instruction and response
- Educational value and helpfulness
- Reflection of human preferences and real-world applicability

Scoring guidelines:

- 1-2: Poor quality, incorrect, unhelpful, or completely misaligned
- 3-4: Below average, partially correct but lacks clarity or completeness
- 5-6: Average quality, generally correct but could be improved
- 7-8: Good quality, accurate, clear, and helpful
- 9-10: Excellent quality, comprehensive, insightful, and highly valuable

Please respond with ONLY a single number (1-10) representing the quality score.

## D EXAMPLES FROM PIKA DATASET

To illustrate the application of our evaluation framework, we present two representative examples from the PiKa dataset with their corresponding instruction-response pairs.

### D.1 EXAMPLE 1: COMPLEX SCIENTIFIC ANALYSIS

**Instruction:**

> Provide an in-depth analysis of non-lethal whale research methods used in the Southern Ocean, including their effectiveness compared to lethal research practices. Discuss the ethical, ecological, and scientific advantages of these methods, and provide examples of successful non-lethal whale studies. Additionally, examine Japan's whaling practices from a conservation perspective and explain how international cooperation, particularly through the International Whaling Commission, can promote sustainable whale management.

**Response:**

> **Non-Lethal Whale Research Methods in the Southern Ocean: An In-Depth Analysis**
>
> Whale research in the Southern Ocean has evolved significantly, with non-lethal methods becoming increasingly effective alternatives to lethal whaling practices. Non-lethal techniques provide valuable scientific data while preserving whale pop-

ulations and maintaining ecological balance. These methods are widely recognized for their ethical, ecological, and scientific advantages over lethal research practices.

**Non-Lethal Research Methods**

Several non-lethal approaches have been successfully implemented in whale research, demonstrating their effectiveness in studying whale populations, behavior, and health:

**1. Biopsy Sampling**

- Small tissue samples are obtained using a specialized dart system.
- Provides data on genetics, hormone levels, pollutants, and diet.
- Less invasive and does not harm the whale.

**2. Photo Identification**

- Researchers track individual whales using unique markings and patterns on their bodies.
- Allows long-term monitoring of population dynamics and migration patterns.
- Successfully used to estimate population sizes and reproductive rates.

**3. Acoustic Monitoring**

- Hydrophones record whale vocalizations to study communication and population distribution.
- Helps assess the impact of noise pollution on whale behavior.
- Provides real-time data on species presence and movement.

**4. Satellite Tagging**

- Non-invasive tags are attached to whales to track migration patterns and feeding behavior.
- Offers insights into habitat use and climate change impacts on whale movements.
- Long-term tracking improves conservation strategies.

**5. Environmental DNA (eDNA) Analysis**

- Water samples are analyzed for genetic material shed by whales.
- Enables species identification and population assessments without direct contact.
- Emerging as a highly effective and non-invasive research tool.

**Effectiveness Compared to Lethal Research Practices**

Non-lethal methods provide comprehensive scientific data without harming whale populations. Lethal research, historically justified for studying whale age, diet, and reproductive health, has been largely discredited due to advancements in non-lethal techniques. The effectiveness of non-lethal methods is evident in their ability to:

- Yield high-quality genetic, ecological, and behavioral data.
- Allow long-term monitoring of individual whales.
- Avoid disrupting population stability and ecosystem balance.

**Ethical, Ecological, and Scientific Advantages**

*Ethical Benefits*

- Eliminates unnecessary harm and suffering.
- Aligns with global conservation and animal welfare principles.
- Enhances public support for whale research and conservation.

*Ecological Benefits*

- Preserves whale populations, ensuring their role in marine ecosystems.
- Prevents disruptions in food chains and biodiversity.
- Supports sustainable ocean management efforts.

*Scientific Benefits*

- Provides more reliable and replicable data over extended periods.
- Facilitates international collaboration in whale conservation.
- Advances knowledge of whale behavior and adaptation to environmental changes.

**Successful Non-Lethal Whale Studies**

Several studies highlight the success of non-lethal research in the Southern Ocean:

- **The Antarctic Humpback Whale Project** (AHWP): Uses photo identification and satellite tagging to monitor whale populations.
- **Southern Ocean Research Partnership (SORP):** A multinational initiative employing acoustic monitoring and biopsy sampling.
- **Australian Antarctic Division's Whale Research Program:** Uses eDNA analysis and aerial surveys to assess whale distribution.

**Japan's Whaling Practices from a Conservation Perspective**

Japan has historically conducted whaling under the guise of scientific research, particularly in the Southern Ocean. Despite the 1986 International Whaling Commission (IWC) moratorium on commercial whaling, Japan has continued whaling operations through loopholes in the IWC framework.

*Conservation Concerns*

- **Population Decline:** Targeting endangered species, such as minke and fin whales, threatens population recovery.
- **Ecosystem Disruption:** Whales play a crucial role in nutrient cycling, and their removal affects marine biodiversity.
- **Questionable Scientific Justification:** Most data obtained through lethal whaling can be collected using non-lethal methods.

**International Cooperation and the Role of the IWC**

The **International Whaling Commission (IWC)** plays a vital role in promoting sustainable whale management through international cooperation. Key strategies include:

**1. Strengthening International Regulations**

- Enforcing the moratorium on commercial whaling.
- Closing loopholes that allow scientific whaling to continue.
- Implementing stricter penalties for non-compliance.

**2. Encouraging Non-Lethal Research**

- Providing funding and support for non-lethal whale studies.
- Facilitating data-sharing initiatives among member nations.
- Promoting ethical research practices.

**3. Diplomatic and Economic Pressure**

- Encouraging Japan and other whaling nations to transition to whale-watching industries.
- Leveraging trade agreements to incentivize compliance with conservation efforts.
- Engaging in diplomatic negotiations to phase out unnecessary whaling.

**4. Public Awareness and Advocacy**

- Increasing global awareness of whale conservation issues.
- Supporting NGOs and scientific communities advocating for non-lethal research.
- Encouraging responsible ecotourism as an alternative to whaling.

**Conclusion**

Non-lethal whale research methods in the Southern Ocean have proven to be highly effective, offering ethical, ecological, and scientific advantages over lethal practices. Japan's continued whaling poses significant conservation challenges,

but international cooperation through the IWC and other global initiatives can promote sustainable whale management. By prioritizing non-lethal research and enforcing conservation regulations, the global community can ensure the protection and long-term survival of whale populations.

## D.2 EXAMPLE 2: MEDICAL INFORMATION REQUEST

**Instruction:**

Explain the common causes of eye twitching and provide practical strategies for treatment and prevention, including relaxation techniques, dietary adjustments, and medical interventions. Additionally, discuss rare cases where eye twitching persists despite treatment and when surgery may be considered as a last resort.

**Response:**

**Common Causes of Eye Twitching**

Eye twitching, also known as **myokymia**, is an involuntary spasm of the eyelid muscles. It is usually benign and temporary, but persistent twitching can be bothersome. Common causes include:

1. **Fatigue and Lack of Sleep** – Insufficient rest can lead to muscle spasms, including twitching in the eyelid.
2. **Stress and Anxiety** – Emotional strain can trigger involuntary muscle movements.
3. **Caffeine and Alcohol Consumption** – Excessive intake of stimulants can overexcite the nervous system, leading to twitching.
4. **Eye Strain** – Prolonged screen time or reading without breaks can fatigue the eye muscles.
5. **Dry Eyes** – A lack of moisture can irritate the eye and contribute to twitching.
6. **Nutritional Deficiencies** – Lack of essential nutrients, such as magnesium, potassium, or vitamin B12, can cause muscle spasms.
7. **Allergies** – Eye irritation from allergens can lead to twitching.
8. **Medication Side Effects** – Some medications, including those for epilepsy and mental health conditions, may cause eye twitching.

**Treatment and Prevention Strategies**

*1. Relaxation Techniques*

- **Stress Management:** Engage in activities like meditation, deep breathing, or yoga to reduce stress.
- **Adequate Sleep:** Ensure you get 7-9 hours of sleep per night to allow your body to recover.
- **Eye Rest:** Follow the **20-20-20 rule**—every 20 minutes, look at something 20 feet away for 20 seconds to reduce eye strain.

*2. Dietary Adjustments*

- **Reduce Caffeine and Alcohol:** Cutting back on coffee, tea, and alcohol can help minimize twitching.
- **Increase Magnesium Intake:** Magnesium-rich foods (nuts, spinach, bananas) may help prevent muscle spasms.
- **Stay Hydrated:** Dehydration can contribute to twitching, so drink plenty of water.
- **Consume Enough Vitamin B12 and Potassium:** These nutrients support nerve function and muscle health.

*3. Medical Interventions*

- **Artificial Tears:** If dry eyes are a contributing factor, lubricating eye drops can help.

- **Corrective Lenses:** If eye strain is due to vision problems, glasses or contacts may be necessary.
- **Botox Injections:** In cases of chronic twitching, botulinum toxin can temporarily relieve symptoms by relaxing the muscles.
- **Medication Adjustments:** If twitching is a side effect of a prescribed drug, consult your doctor about alternatives.

**Persistent Eye Twitching and Surgical Considerations**

In rare cases, **eye twitching may persist despite treatment**, potentially indicating an underlying neurological disorder such as **benign essential blepharospasm** or **hemifacial spasm**. If conservative treatments fail, surgery may be considered:

- **Myectomy:** A surgical procedure that removes some of the muscles responsible for twitching.
- **Microvascular Decompression (for Hemifacial Spasm):** A neurosurgical procedure that relieves pressure on facial nerves.

**When to See a Doctor**

Seek medical attention if:

- Twitching persists for **more than a few weeks**.
- It affects **other parts of the face**.
- There is **difficulty opening the eye**.
- You experience **vision changes or pain**.

**Conclusion**

While most cases of eye twitching are harmless and temporary, addressing lifestyle factors such as sleep, stress, and nutrition can help alleviate symptoms. If twitching persists despite treatment, medical interventions—including Botox or, in rare cases, surgery—may be necessary. Consulting a healthcare provider is essential for persistent or severe cases to rule out underlying conditions.

# E  USE OF LARGE LANGUAGE MODELS IN PAPER WRITING

We used LLMs to assist with language polishing and minor formatting of the paper. No LLMs were involved in research ideation.

# ADDITIONAL ANALYSES AND CLARIFICATIONS

## E.1 EFFECT OF PROMPT DIFFICULTY

PiKa adopts a difficulty-driven construction strategy. To examine whether prompt difficulty *causally* improves alignment performance, we apply a down-quality reconstruction procedure to PiKa prompts and regenerate lower-difficulty variants using the guided template shown below. Table 4 shows consistent monotonic improvements as difficulty increases.

---

**System Prompt:** You are an expert evaluator for assessing the difficulty level of instructions/questions. Please evaluate the given instruction on a scale of 1–10 based on:

- Complexity of the task or question
- Level of domain knowledge required
- Cognitive load needed to understand and process
- Technical depth and sophistication

Scoring guidelines:

- 1–2: Very basic, elementary level knowledge
- 3–4: Intermediate, requires some specialized knowledge
- 5–6: Advanced, requires solid domain expertise
- 7–8: Expert level, requires deep specialized knowledge
- 9–10: Cutting-edge, requires highly specialized or research-level expertise

Now, instead of giving a score, please **rewrite the given instruction** into a new version that would have a difficulty score of **1–2** according to the above scale.
Guidelines:

- Keep the topic and domain consistent
- Simplify the reasoning and remove complexity
- Avoid multi-step or abstract reasoning

Output strictly in JSON: { "instruction": "<rewritten easy instruction>" }

---

As shown above, PiKa exhibits substantially higher prompt difficulty than Magpie (average 7.39 vs. 2.65). To more directly test the causal effect of difficulty, we reconstruct lower-quality prompts and regenerate responses using the above template. Each level reduces prompt difficulty while holding other factors fixed. The "mid" variant is derived from the vanilla prompt, and the "low" variant is derived from the "mid" prompt. Difficulty is re-evaluated with GPT-4o.

$$
\begin{aligned}
P'_{\text{new}} &= \text{LLM}(P_{\text{vanilla}}, \text{score\_guide}), \\
R_{\text{new}} &= \text{LLM}(P'_{\text{new}}).
\end{aligned}
\tag{10}
$$

The results in Table 4 confirm that higher prompt difficulty leads to stronger alignment gradients.

Table 4: Difficulty ablation (10k samples).

| Model (10k) | Difficulty | LC (%) | WR (%) |
|---|---|---|---|
| LLaMA-3-8B-Magpie-Pro | 2.65 | 15.42 | 16.89 |
| LLaMA-3-8B-PiKa (low) | 2.91 | 21.86 | 14.95 |
| LLaMA-3-8B-PiKa (mid) | 3.64 | 24.36 | 17.84 |
| **LLaMA-3-8B-PiKa** | **7.39** | **31.01** | **30.32** |

## E.2 EFFECT OF PERSONA CONDITIONING

Persona conditioning encourages generation of more specialized, higher-difficulty, and less redundant prompts. Removing persona information substantially reduces both difficulty and diversity. Results are shown in Table 5.

Table 5: Persona ablation (10k samples). Mean MND (Minimum Neighbor Distance) measures instruction redundancy (higher is better).

| Model (10k) | Persona | Difficulty | Mean MND | LC (%) | WR (%) |
|---|---|---|---|---|---|
| LLaMA-3-8B-PiKa | none | 3.11 | 0.0234 | 13.84 | 15.53 |
| **LLaMA-3-8B-PiKa** | enabled | **7.39** | **0.497** | **31.01** | **30.32** |

## E.3 PERSONA DOMAIN DISTRIBUTION

We analyze persona-domain coverage across the PiKa-30k dataset. Table 6 shows that personas largely fall within broad, widely applicable domains rather than narrow or niche specialties.

Table 6: Top 10 persona-domain distribution in PiKa (percentage).

| Domain | Percentage |
|---|---|
| General | 36.3% |
| Medicine | 16.0% |
| Biology | 13.3% |
| History | 12.5% |
| Engineering | 6.8% |
| Environmental Science | 6.5% |
| Education | 5.8% |
| Psychology | 4.3% |
| Computer Science | 4.0% |

To assess robustness to domain composition, we train three PiKa-10k subsets sampled with different seeds. Variance across runs is minimal.

Table 7: Robustness across different random samples of PiKa (10k).

| Experiment | LC (%) | WR (%) |
|---|---|---|
| 1 | 31.01 | 30.32 |
| 2 | 30.09 | 30.17 |
| 3 | 30.45 | 30.06 |

## E.4 COMPARISON WITH MOAA: CONTROLLED EVALUATION

We compare PiKa to MoAA under matched evaluation settings. For LLaMA-3.1-8B-Instruct, we directly evaluate the MoAA-SFT checkpoint released by Wang et al. (2025), and compare it with our PiKa-30k fine-tuned version of the same model. In contrast, for the open-source LLaMA-3-8B-Base backbone, we train both PiKa and MoAA variants ourselves to ensure a controlled comparison.

## E.5 EVALUATION BIAS ACROSS DIFFERENT JUDGES

To examine whether using GPT-4o as an evaluator introduces systematic bias, we further evaluate PiKa, LLaMA-3-Instruct, and Magpie-Pro using two independent judges: **GPT-5** and **GPT-4.1-mini**.

Table 8: Comparison on LLaMA-3-1-8B-Instruct using PiKa-30k vs. MoAA-SFT.

| Model | LC (%) | WR (%) | SD | N | Avg Len |
|---|---|---|---|---|---|
| **PiKa-SFT** | **42.94** | 38.46 | 1.63 | 805 | 1790 |
| MoAA-SFT | 38.16 | 38.33 | 1.65 | 805 | 2235 |

Table 9: Open-source comparison on LLaMA-3-8B-Base.

| Model | LC (%) | WR (%) |
|---|---|---|
| LLaMA-3-8B-Magpie-Pro | 24.06 | 28.60 |
| LLaMA-3-8B-MoAA | 28.16 | 27.17 |
| **LLaMA-3-8B-PiKa** | **32.82** | **30.56** |

Results in Table 10 show that PiKa consistently outperforms baselines across all evaluators, indicating robustness to choice of judge.

Table 10: Evaluation across different judges.

| Judge | Model | LC (%) | WR (%) | SD | N | Avg Len |
|---|---|---|---|---|---|---|
| | LLaMA-3-8B-PiKa | 19.28 | 19.00 | 1.35 | 805 | 1927 |
| GPT-5 | LLaMA-3-8B-Instruct | 15.95 | 16.56 | 1.28 | 805 | 1949 |
| | LLaMA-3-8B-Magpie-Pro | 9.51 | 14.55 | 1.19 | 805 | 2461 |
| | LLaMA-3-8B-PiKa | 29.02 | 27.55 | 1.52 | 805 | 1927 |
| GPT-4.1-mini | LLaMA-3-8B-Instruct | 25.44 | 25.53 | 1.45 | 805 | 1949 |
| | LLaMA-3-8B-Magpie-Pro | 20.17 | 28.54 | 1.53 | 805 | 2461 |

## E.6 LENGTH ANALYSIS ACROSS 5,000 PROMPTS

We analyze 5,000 prompts, each paired with five candidate responses, to examine whether verbosity bias affects reward-model preferences. Reported lengths correspond to **character lengths** (approximately 1,300 tokens). Across all prompts, the chosen responses are on average only **7.5%** longer than the rejected ones (5,343 vs. 4,969 characters). Moreover, only **30.8%** of prompts select the longest candidate as the preferred response, and the global correlation between response length and reward score remains mild ($r = 0.31$). These observations collectively indicate that length is not the dominant factor driving reward-model decisions.

Table 11: Length analysis across 5,000 prompts.

| Metric | Chosen Response | Rejected Response | $\Delta\%$ | Longest Selected (%) | Corr (r) |
|---|---|---|---|---|---|
| Length (chars) | 5,343 | 4,969 | +7.5% | 30.8% | 0.31 |

