# OpenReview forum: "PIKA: Expert-Level Synthetic Datasets for Post-Training Alignment from Scratch"
_ICLR.cc/2026/Conference — Submitted to ICLR 2026_

### Official Review · Reviewer_Uvpd · 2025-10-31

**Soundness:** 2
**Presentation:** 3
**Contribution:** 3
**Rating:** 4
**Confidence:** 2

**Summary:**

This paper presents a highly impactful finding that directly challenges the "scale-is-all-you-need" paradigm for LLM alignment. The core contribution is PIKA, a remarkably small (only 30k SFT samples) yet exceptionally high-quality synthetic dataset.

**Strengths:**

PIKA Dataset Construction: The paper proposes a clean, three-step pipeline (Figure 1) to generate expert-level data16:
(1) Expert-Level Prompt Generation: It uses a persona-based approach, sampling complex, expert personas (e.g., from biology, law) from PersonaHub and prompting GPT-4O to generate knowledge-intensive instructions.
(2) Multi-Path Response Generation: For each instruction, the LLM generates $k$ candidate responses18.
(3) Reward-Model-Guided Selection: A strong reward model (RM) is used to score all responses. The PiKa-SFT dataset retains only the (instruction, highest-scoring response) pair.

**Weaknesses:**

PIKA's "expert-level" quality is not created from scratch; it is distilled. The framework relies on GPT-4O to generate high-quality instructions and a state-of-the-art Reward Model to filter and select the best responses . This means PIKA's success is fundamentally dependent on the capabilities of these more powerful models, which were themselves trained on massive-scale data.

This process introduces strong, unanalyzed stylistic biases. For example, the dataset's average response length is massive (5,365 tokens), suggesting the pipeline strongly rewards verbosity. The paper does not analyze whether this "longer is better" bias is truly optimal or just an artifact of the specific reward model chosen.

**Questions:**

plz check the weakness

---

> ### Author Response · Authors · 2025-11-16
> **Rebuttal**
>
> Thank you for your time and effort in reviewing our work. We would like to address the concern you raised as follows:
>
> ---
>
> > “PIKA's ‘expert-level’ quality is not created from scratch; it is distilled. The framework relies on GPT-4o to generate high-quality instructions and a state-of-the-art Reward Model to filter and select the best responses. This means PIKA's success is fundamentally dependent on the capabilities of these more powerful models, which were themselves trained on massive-scale data.”
>
> Thank you for the comment. We would like to clarify that the phrase “from scratch” in our title refers to the **post-training process**, rather than implying that the dataset itself is human-constructed from scratch. Specifically, PiKa fine-tunes base models (for example, LLaMA-3-8B-Base and Qwen2.5-Base) **without any prior instruction tuning or human-curated datasets**, making the entire post-training alignment stage more open and reproducible.
>
> As noted in the Abstract and Introduction, PiKa belongs to the **distillation-based synthetic data** paradigm, leveraging AI feedback that has been widely adopted in both research and industry for efficient post-training [1, 2, 3]. Our goal is to demonstrate that when such synthetic data is properly guided, filtered, and curated, it can achieve **high data efficiency** in aligning base models and significantly reduce the reliance on proprietary datasets.
>
> [1] CodecLM: Aligning Language Models with Tailored Synthetic Data (By Google)
>
> [2] Textbooks Are All You Need (By Microsoft)
>
> [3] A Synthetic Dataset for Personal Attribute Inference (NeurIPS 2024)
>
> ---
>
> > “This process introduces strong, unanalyzed stylistic biases. For example, the dataset's average response length is massive (5,365 tokens), suggesting the pipeline strongly rewards verbosity. The paper does not analyze whether this ‘longer-is-better’ bias is truly optimal or just an artifact of the specific reward model chosen.”
>
> Thank you for this question. We would like to clarify that the reported number 5,365 refers to **character length** rather than token length (a typo in the previous text, which has been corrected in the new version). This corresponds to approximately **1,300 tokens**, which is not indicative of excessive verbosity. Examples can be found in Appendix D.
>
> To further examine whether verbosity bias is present, we analyzed 5,000 prompts (each associated with 5 candidate responses). We observe that the **average chosen response** is only **7.5%longer** than rejected ones (5,343 vs. 4,969 characters), and only **30.8%** of prompts select the longest response as the preferred one. These results indicate that length is **not the dominant factor** determining the reward model's decision.
>
> We additionally computed the correlation between response length and reward score (r = 0.31), suggesting only a mild global tendency toward verbosity. Taken together, these findings imply that the reward model focuses primarily on **content quality rather than length**. We also note that the reward model used in PiKa is currently ranked **#Top 1** on RewardBench and supports up to 16,384 input tokens, ensuring that our inputs remain within the model's recommended operational range. Prior work has similarly shown the effectiveness of high-quality reward models pre-trained with massive human-preference datasets in reflecting human preference judgments [4, 5, 6].
>
> | Metric | Chosen | Rejected | Δ% | Longest Selected (%) | Corr(r) |
> |--------|---------|-----------|------|----------------------|----------|
> | Length (chars) | 5,343 | 4,969 | +7.5 percent | 30.8 percent | 0.31 |
>
> [4] Self-Play Preference Optimization for Language Model Alignment (ICLR 2025)
> [5] SimPO: Simple Preference Optimization with a Reference-Free Reward (NeurIPS 2025)
> [6] Magpie: Alignment Data Synthesis from Scratch by Prompting Aligned LLMs with Nothing (ICLR 2025)
>
> ---
>
> We thank you again for your valuable feedback and would greatly appreciate it if you could reconsider your evaluation should our responses address your concerns.

---

> > ### Comment · Reviewer_Uvpd · 2025-11-16
> >
> > Thanks for the reply. My main issue was the 5,365-token average length, which seemed like a massive bias.
> >
> > The authors clarified this was a character count, not a token count. This was a critical typo, and it resolves my biggest objection.
> >
> > I'm still not a huge fan of the "distillation" approach, but the paper's core data efficiency claim (30k vs 10M+) looks solid now. I was on the fence, but since my main problem is fixed, I'm raising my score.

---

> > > ### Author Response · Authors · 2025-11-16
> > >
> > > Thank you very much for raising your score! We are glad that the clarification resolved your concerns, and we appreciate your recognition of our work’s data-efficiency contribution. This supports our core claim, and we hope the work can help lower the entry barrier for the whole open-source post-training research!

---

### Official Review · Reviewer_Et4v · 2025-10-31

**Soundness:** 3
**Presentation:** 3
**Contribution:** 2
**Rating:** 4
**Confidence:** 3

**Summary:**

The authors introduce PiKa, a synthetic data suite for post-training alignment. It explores data generation means by persona-driven prompts and filtered by a reward model. The contributions come in two parts: (1) PiKa-SFT with ~30K instruction-response pairs, and (2) a matching preference dataset ~30K chosen vs rejected responses for DPO tuning. The result shows the models tuned on PiKa beat those trained on much larger public datasets on AlpacaEval 2.0 and Arena-Hard.

**Strengths:**

1. With just 30K SFT pairs, PiKa beats much larger public datasets.

2. The generation scheme is simple, reproducible.

**Weaknesses:**

1. DPO pairs are constructed by taking top vs bottom responses under a single reward model. Models may learn the reward model rather than robust human preferences.

2. Headline claims hinge on AlpacaEval 2.0 and Arena-Hard; broader real-world tasks like safety, long-context aren’t reported hear.

**Questions:**

Q1: The baselines include multi-turn conversational datasets (e.g., ShareGPT, WildChat), if I am understanding correctly, Pika seems to be more single-turn and expert-focused. Could the performance gains be partly attributed to a closer domain match between Pika's data style and the evaluation benchmarks (AlpacaEval, Arena-Hard), rather than inherent superiority? How would a Pika-trained model perform on a multi-turn conversational benchmark? It would be better if authors could discuss more about data type differences in both train and test sets.

Q2: The paper uses GPT-4o for instruction generation and as the "LLM-as-judge" for benchmark evaluation. Could this create a potential bias where the model is being evaluated by the same system that helped create its training data?

Q3: Pika's responses are an order of magnitude longer than those in other datasets. Given the known bias of LLM judges towards longer responses, to what degree can the superior win rates be attributed to response length rather than fundamental improvements in instruction-following quality?

---

> ### Author Response · Authors · 2025-11-16
> **Rebuttal (Page 1)**
>
> Thank you for your time and effort in reviewing our work. We would like to address the questions you raised as follows:
>
> ---
>
> > “DPO pairs are constructed by taking top vs bottom responses under a single reward model. Models may learn the reward model rather than robust human preferences.”
>
> Thank you for the comment. We would like to clarify that different reward models exhibit **different reward score distributions** even on the same response pairs [4], and combining multiple reward models without additional training is generally not feasible or meaningful. Constructing pairs by selecting the top and bottom responses under a **single strong reward model** is a standard and widely adopted practice in both research and industry for RLHF and RLAIF pipelines [1–3]. This approach has repeatedly been shown to be both effective and cost-efficient.
>
> Reward models used in these pipelines are trained on **large-scale, human-annotated preference datasets**, and therefore provide a strong and stable human-aligned evaluation signal. In PiKa, we use Skywork-Reward-V2-LLaMA-3.1-8B, which ranks at the top of RewardBench. This choice ensures reliable preference ranking while keeping the data construction process reproducible for the open-source community.
>
> [1] Self-Play Preference Optimization for Language Model Alignment (ICLR 2025)
>
> [2] SimPO: Simple Preference Optimization with a Reference-Free Reward (NeurIPS 2025)
>
> [3] Magpie: Alignment Data Synthesis from Scratch by Prompting Aligned LLMs with Nothing (ICLR 2025)
>
> [4] How to Evaluate Reward Models for RLHF (ICLR’25)
>
> ---
>
> > “Headline claims hinge on AlpacaEval 2.0 and Arena-Hard; broader real-world tasks like safety, long-context aren’t reported here.”
>
> Thank you for pointing this out. In fact, we have already included a wide range of **real-world task evaluations** in Table 3 (page 8), covering aspects such as reasoning, factuality, robustness, etc.. These include MMLU, ARC-Challenge, HellaSwag, TruthfulQA, WinoGrande, and GSM8K.
>
> Our focus in this paper is **post-training alignment** for general instruction-following, which aligns with the evaluation setups used in recent alignment studies [1–3]. Long-context reasoning is indeed important, but it is typically addressed through architectural modifications or pretraining strategies, rather than SFT-only alignment [5-7], and therefore falls outside this paper’s scope.
>
> [5] Effective Long‑Context Scaling of Foundation Models (NAACL’24)
>
> [6] A Survey of Techniques to Extend the Context Length in Large Language Models (IJCAI’24)
>
> [7] LongRecipe: Recipe for Efficient Long Context Generalization in Large Language Models
> (ACL’25)
>
>
> ---
>
> > “The baselines include multi-turn conversational datasets (e.g., ShareGPT, WildChat). If I am understanding correctly, PiKa seems to be more single-turn and expert-focused. Could the performance gains be partly attributed to a closer domain match between PiKa’s data style and the evaluation benchmarks (AlpacaEval, Arena-Hard), rather than inherent superiority? How would a PiKa-trained model perform on a multi-turn conversational benchmark?”
>
> Thank you for the comment. Multi-turn datasets such as ShareGPT and WildChat are included only to provide a comprehensive baseline comparison. Recent SoTA alignment datasets, such as Magpie-Pro, and most preference-optimization work also adopt the **single-turn setting**, as it simplifies preference labeling and reflects how many real-world interactions are evaluated [1–3].
>
> Importantly, the evaluation benchmarks are **not** more aligned with PiKa’s data style. AlpacaEval 2.0 is a general-purpose chat benchmark with relatively low average difficulty (approximately 3.20), while PiKa focuses on expert-level task prompts. Arena-Hard contains primarily math and code questions, which are not directly emphasized in PiKa. These differences indicate that PiKa’s performance gains arise from **data quality rather than domain overlap**.

---

> ### Author Response · Authors · 2025-11-16
> **Rebuttal (Page 2)**
>
> > “The paper uses GPT-4o for instruction generation and as the ‘LLM-as-judge’ for benchmark evaluation. Could this create a potential bias where the model is being evaluated by the same system that helped create its training data?”
>
> Thank you for raising this concern. We use GPT-4o as one of the judges because AlpacaEval 2.0 reports that GPT-4-level evaluators achieve **over 68 percent agreement with human judgments**, higher than average human annotators (65.7 percent), and have the lowest bias rates among automatic evaluators.
>
> To further address potential evaluation bias, we conducted additional experiments using **GPT-5** and **GPT-4.1-mini** as independent judges. As shown below, PiKa consistently outperforms both LLaMA-3-Instruct and Magpie-Pro across all evaluators, demonstrating robustness against evaluator bias.  (Have included in **Appendix E.5**）
>
> | Judge Model | Model Variant | LC | WR | SD | n_total | Avg Len |
> |-------------|---------------------------|------:|------:|------:|--------:|--------:|
> | GPT-5       | LLaMA-3-8B-PiKa           | 19.28 | 19.00 | 1.35 | 805 | 1927 |
> |             | LLaMA-3-8B-Instruct       | 15.95 | 16.56 | 1.28 | 805 | 1949 |
> |             | LLaMA-3-8B-Magpie-Pro     |  9.51 | 14.55 | 1.19 | 805 | 2461 |
> | GPT-4.1-mini | LLaMA-3-8B-PiKa          | 29.02 | 27.55 | 1.52 | 805 | 1927 |
> |             | LLaMA-3-8B-Instruct       | 25.44 | 25.53 | 1.45 | 805 | 1949 |
> |             | LLaMA-3-8B-Magpie-Pro     | 20.17 | 28.54 | 1.53 | 805 | 2461 |
>
> The consistency across judges supports the validity of PiKa’s improvements.
>
> ---
>
> > “PiKa’s responses are an order of magnitude longer than those in other datasets. Given the known bias of LLM judges toward longer responses, to what degree can the superior win rates be attributed to response length rather than fundamental improvements in instruction-following quality?”
>
> Thank you for raising this question. We clarify that this statistic refers to **dataset sample lengths**, which are naturally longer because expert-level prompts require more detailed explanations to distill knowledge into the model. However, model output lengths during evaluation are much shorter.
>
> More importantly, AlpacaEval 2.0 uses **Length-Controlled Win Rate (LC)** to normalize for length effects. As shown above, LLaMA-3-8B-PiKa actually produces **shorter responses** on average (1,927 tokens) than the baselines, yet achieves the highest LC scores. This demonstrates that PiKa’s improvements arise from **better alignment and response quality**, not verbosity.
>
> ---
>
> We thank you again for your valuable feedback and would greatly appreciate it if you could reconsider your evaluation should our responses address your concerns.

---

> ### Author Response · Authors · 2025-11-28
>
> Dear Reviewer Et4v,
>
> Regarding your major concerns about real-world task evaluation, we have included relevant experiments in our paper. For the bias concern of LLM judges, we have used multi-judge evaluation with GPT-5 and GPT-4.1-mini to mitigate potential biases. More detailed content and analysis can be found in our previous messages.
>
> We hope that our responses have adequately addressed your concerns. As the deadline for this discussion phase is approaching, we warmly welcome any further discussion or clarifications regarding additional concerns you may have.
>
> Best,
>
> Authors

---

### Official Review · Reviewer_qzhQ · 2025-10-31

**Soundness:** 3
**Presentation:** 3
**Contribution:** 1
**Rating:** 2
**Confidence:** 4

**Summary:**

PiKa is a compact, expert-level synthetic dataset for aligning LLMs that dramatically improves data efficiency in instruction tuning. Built using persona-driven instruction generation and reward-model filtering with GPT-4o, PiKa produces 30k high-difficulty, domain-rich instruction-response pairs and 30k preference examples. Models fine-tuned on PiKa outperform those trained on datasets over ten times larger, even surpassing proprietary instruction-tuned models like Llama-3-8B-Instruct on key alignment benchmarks. This shows that careful synthesis of expert-level data can replace massive corpora.

**Strengths:**

- The proposed pipeline is effective at improving two popular models in terms of alignment while minimally impact other capabilities.
- The methodology as well as data curation process are well presented and easy to follow.
- The clarity and structure of the paper are commendable.

**Weaknesses:**

```The method's performance is struggling```
Constructing synthetic data for alignment purposes is not new. The paper introduces another new way of constructing data. However, the evidence from the performance improvements are not strong enough to justify a publication. Magpie came out over a year ago and since then there has been plenty of work that's able to improve AlpacaEval and ArenaHard to new heights. For example, Wang et al. [1] improves llama3-8b to have 57.23 for AlpacaEval 2 and 48.3 for Arena-Hard with <70k data samples. This is much higher than Pika with 32.82 on AlpacaEval2 and 33.5 on Arenahard. The novelty of this data construction pipeline is hard to justify if it struggles to beat works in 2025. I seriously doubt the scalability of such methods, unless authors can provide other evidences.

[1] Improving Model Alignment Through Collective Intelligence of Open-Source LLMS

```Why does difficulty help is not clear```
The link between dataset “difficulty” and alignment improvement could be further substantiated with ablation or causal evidence rather than correlation.

```Does persona actually help?```
While persona-based generation is compelling, the paper could provide more analysis of persona diversity and domain distribution, as these may critically influence performance. It is also unclear how helpful are those persona in terms of quality. It seems like an ablation without persona may be needed.

**Questions:**

- Could the authors provide more insight into the cost-efficiency tradeoff—for example, how expensive it was to generate, score, and filter 30k examples compared to traditional human-annotated datasets?
- Would the PiKa approach scale effectively to larger datasets, or do the authors expect diminishing returns due to redundancy or computational overhead?

---

> ### Author Response · Authors · 2025-11-16
> **Rebuttal (Page 1)**
>
> Thank you for your time and effort in reviewing our work. We would like to address the questions you raised as follows:
>
> ---
>
> > “The method's performance is struggling. Constructing synthetic data for alignment purposes is not new. The paper introduces another new way of constructing data. However, the evidence from the performance improvements are not strong enough to justify a publication. Magpie came out over a year ago and since then there has been plenty of work that's able to improve AlpacaEval and ArenaHard to new heights. For example, Wang et al. [1] improves LLaMA-3-8B to have 57.23 for AlpacaEval 2 and 48.3 for Arena-Hard with <70k data samples. This is much higher than PiKa with 32.82 on AlpacaEval 2 and 33.5 on Arena-Hard. The novelty of this data construction pipeline is hard to justify if it struggles to beat works in 2025. I seriously doubt the scalability of such methods, unless authors can provide other evidences.”
>
> Thank you for the comment. The comparison to Wang et al. [1] involves a **fundamentally different** model initialization setup. Wang et al. fine-tune LLaMA-3.1-8B-Instruct, which is a proprietary instruction-tuned model already trained with large closed-source post-training data. In contrast, our experiments use LLaMA-3-8B-Base, a pretrained model without any instruction tuning. This distinction is essential: PiKa aims to demonstrate effective alignment from base models in a fully open-source setting. The goal of Wang et al. is to further optimize an already instruction-tuned model, whereas ours is to establish a data-efficient post-training pipeline starting from scratch.
>
> To ensure fairness, we additionally fine-tuned LLaMA-3.1-8B-Instruct using PiKa (30k examples) under our standard configuration and compared it directly with the open checkpoint from Wang et al. (togethercomputer/Llama-3.1-8B-Instruct-MoAA-SFT). All evaluations were conducted using the GPT-4o API to ensure consistent and up-to-date judgment (We note that Wang et al. [1] relied on the older gpt-4-1106-preview evaluator, which has been reported to behave differently from newer GPT-4o variants, and relative analysis can be found in [2] ).  (Have included in **Appendix E.4**）
>
> #### AlpacaEval 2.0
> | Model | LC | WR | SE | n_total | Avg Length |
> |-------|----|----|----|----------|-------------|
> | LLaMA-3.1-8B-Instruct + PiKa-SFT | 42.94 | 38.46 | 1.63 | 805 | 1790 |
> | LLaMA-3.1-8B-Instruct + MoAA SFT [1] | 38.16 | 38.33 | 1.65 | 805 | 2235 |
>
> PiKa achieves a +4.78 LC improvement while using fewer examples (30k vs 66k), demonstrating **better data efficiency**.
>
> We also trained LLaMA-3-8B-Base using MoAA-SFT under the same setup for an open comparison.
>
> | Model | LC | WR |
> |--------|----|----|
> | LLaMA-3-8B-Magpie-Pro | 24.06 | 28.60 |
> | LLaMA-3-8B-MoAA | 28.16 | 27.17 |
> | LLaMA-3-8B-PiKa | 32.82 | 30.56 |
>
> These results confirm that PiKa remains competitive in both closed-source and open-source settings, balancing **performance and data efficiency**.
>
> ---
>
> > “Why does difficulty help is not clear. The link between dataset ‘difficulty’ and alignment improvement could be further substantiated with ablation or causal evidence rather than correlation.”
>
> Thank you for the comment. Figure 3 illustrates that PiKa exhibits substantially higher prompt difficulty than Magpie (average difficulty 7.39 vs. 2.65). To provide more causal evidence, we conducted a controlled difficulty ablation by applying a down-quality reconstruction procedure to PiKa prompts and regenerating lower-difficulty variants using the guided template described in the updated **Appendix E.1**. Each difficulty tier is produced by reducing the prompt complexity while keeping all other factors constant.
>
>
> | Model (10k data) | Difficulty | LC | WR |
> |------------------|------------|----|----|
> | LLaMA-3-8B-Magpie-Pro | 2.65 | 15.42 | 16.89 |
> | LLaMA-3-8B-PiKa (low) | 2.91 | 21.86 | 14.95 |
> | LLaMA-3-8B-PiKa (mid) | 3.64 | 24.36 | 17.84 |
> | LLaMA-3-8B-PiKa (default) | 7.39 | 31.01 | 30.32 |
>
> Performance improves consistently with higher difficulty, supporting the conclusion that **challenging and expert-level prompts provide stronger alignment signals** beyond simple correlation.

---

> ### Author Response · Authors · 2025-11-16
> **Rebuttal (Page 2)**
>
> > “Does persona actually help? While persona-based generation is compelling, the paper could provide more analysis of persona diversity and domain distribution, as these may critically influence performance.”
>
> Thank you for the question. Persona conditioning helps guide the LLM to generate more specialized, high-difficulty prompts. Removing persona roles makes the data substantially easier and more repetitive. We conducted an ablation removing persona tokens from the prompt template. (Have included in **Appendix E.2**）
>
> | Model (10k data) | Persona Setting | Difficulty | Mean MND | LC | WR |
> |------------------|-----------------|------------|-----------|----|----|
> | LLaMA-3-8B-PiKa (no persona) | none | 3.11 | **0.0234** | 13.84 | 15.53 |
> | LLaMA-3-8B-PiKa (default) | enabled | 7.39 | **0.497** | 31.01 | 30.32 |
>
> Persona guidance significantly improves instruction diversity and reduces redundancy (higher MND indicates more diverse instructions).
>
> To examine persona-domain coverage, we categorize the 30k personas into broad expert domains. Below, we report the Top 10 percentage distribution  (Have included in **Appendix E.3**）:
>
> | Domain | Percentage |
> |--------|------------|
> | General | **36.3%** |
> | Medicine | **16.0%** |
> | Biology | **13.3%** |
> | History | **12.5%** |
> | Engineering | **6.8%** |
> | Environmental Science | **6.5%** |
> | Education | **5.8%** |
> | Psychology | **4.3%** |
> | Computer Science | **4.0%** |
>
> The distribution shows that most personas lie in general or widely applicable expert domains rather than specialized niches, ensuring broad coverage without domain overfitting.
>
> To further test robustness to persona composition, we trained three PiKa-10k models using different random seeds in the 30k dataset:
>
> | Experiment | LC | WR |
> |------------|-----:|-----:|
> | 1 | 31.01 | 30.32 |
> | 2 | 30.09 | 30.17 |
> | 3 | 30.45 | 30.06 |
>
> The small variance confirms that PiKa’s improvements are stable and not sensitive to specific persona-domain mixtures, and that persona conditioning contributes meaningfully to prompt richness and data quality.
>
>
>
> ---
>
> > “Could the authors provide more insight into the cost-efficiency trade-off—for example, how expensive it was to generate, score, and filter 30k examples compared to traditional human-annotated datasets?”
>
> Thank you for the question. Generating and filtering the full PiKa dataset cost approximately 4000 US dollars, dominated by GPT-4o generation and reward-model scoring. Comparable human-annotated datasets of similar scale and difficulty are typically ten to twenty times more expensive and often proprietary. We plan to release PiKa publicly to support reproducible and cost-efficient alignment research.
>
> ---
>
>
> > “Would the PiKa approach scale effectively to larger datasets, or do the authors expect diminishing returns due to redundancy or computational overhead?”
>
> Thank you for the question. As shown in Figure 6, PiKa already achieves strong performance at the 10k scale, and increasing the dataset size beyond 30k yields only marginal gains while incurring substantially higher cost. This pattern is consistent with prior observations that larger SFT datasets tend to introduce redundancy rather than meaningful new supervision. Our focus in this work is precisely to **enable effective post-training with a small, efficient dataset**, rather than relying on large-scale data accumulation. For this reason, we believe future work should prioritize **selective and high-signal data curation** over further scaling, which we view as a more sustainable and impactful direction.
>
>
>
> ---
>
> We thank you again for your valuable feedback and would greatly appreciate it if you could reconsider your evaluation should our responses address your concerns.
>
> [1] Wang et al. (2025). Improving Model Alignment Through Collective Intelligence of Open-Source LLMs.
>
> [2] Eureka: Evaluating and Understanding Large Foundation Models (By Microsoft)

---

> ### Author Response · Authors · 2025-11-28
>
> Dear Reviewer qzhQ,
>
> We have made our best effort to clarify your concerns point by point. Regarding your main concern about the performance comparison with Wang's paper, we have clarified that it represents a fundamentally different setting and provided further analysis in our previous message. **We have also revised the manuscript to cite Wang's paper in the appendix**. Could you please confirm whether our clarifications adequately address your concerns?
>
> Best,
>
> Authors

---

### Official Review · Reviewer_k9nm · 2025-11-01

**Soundness:** 3
**Presentation:** 3
**Contribution:** 2
**Rating:** 2
**Confidence:** 3

**Summary:**

The paper introduces the PiKa dataset for SFT training, which is constructed through an automated pipeline. Various statistics of the dataset are analyzed, and experiments are conducted on multiple base models to demonstrate the effectiveness of the dataset.

**Strengths:**

- The paper is well written and easy to follow.

- Detailed statistics of various datasets are analyzed, and fine-tuning on the proposed dataset achieves better performance than previous ones.

**Weaknesses:**

My main concern lies in the contribution. The proposed construction pipeline is widely used in current synthetic data generation. Could the authors better highlight the unique contributions or distinctive characteristics of the PiKa dataset?

**Questions:**

Please refer the the weakness part.

---

> ### Author Response · Authors · 2025-11-16
> **Rebuttal**
>
> Thank you for your time and effort in reviewing our work. We would like to address the questions you raised as follows:
>
> ---
>
> > “My main concern lies in the contribution. The proposed construction pipeline is widely used in current synthetic data generation. Could the authors better highlight the unique contributions or distinctive characteristics of the PiKa dataset?”
>
> Thank you for the comment. We would like to clarify that the primary area of our submission is “datasets and benchmarks”, and the primary contribution of our work lies in the design and release of PiKa, a synthetic dataset optimized for **data-efficient post-training alignment from scratch**. PiKa contributes to the community in two main ways: (1) it provides a transparent, open-source benchmark dataset for studying data efficiency in alignment, and (2) it offers empirical insights into how **prompt difficulty and prompt–response quality** jointly influence post-training performance. Instead of reusing existing instruction-generation templates, PiKa proposes a **prompt-difficulty-driven construction strategy** that explicitly guides the generation of more challenging and information-rich prompt–response pairs, enabling small-scale but well-curated examples to substantially improve alignment outcomes while reducing redundancy and overall cost.
>
> From a data-composition perspective, existing state-of-the-art datasets such as Magpie-Pro largely contain short and surface-level instructions (e.g., “Write a poem about the ocean”), which, although individually high quality, become highly redundant when scaled to hundreds of thousands of samples. In contrast, PiKa explicitly optimizes for **prompt diversity**, **difficulty**, and **prompt–response quality**, as shown in Figure 3 and Figure 4. This ensures that each example is **high-signal and non-trivial**, enabling models to learn more effectively from fewer examples.
>
> In terms of performance, PiKa is the **first** open-source dataset to outperform proprietary instruction-tuned models on strong backbones such as Llama-3-8B, while using only 30k SFT examples, compared to 300k examples in Magpie-Pro and over 10 million preference and SFT samples used to train the official Llama-3-8B-Instruct model. As shown in Table 1, PiKa-tuned models consistently outperform both base and instruction-tuned counterparts trained on much larger datasets. Table 2 further shows that PiKa generalizes well across backbones (for example, Qwen-2.5 from 0.5B to 7B), achieving competitive results on two widely used alignment benchmarks, AlpacaEval 2.0 and Arena-Hard, both designed to reflect general-purpose chat ability.
>
> ---
>
> Finally, we commit to releasing PiKa upon publication, and we believe it will empower the open-source community to train competitive aligned models from scratch without relying on proprietary data, thereby **lowering the entry barrier** for alignment research and promoting reproducible, data-efficient post-training. We thank you again for your valuable feedback and would greatly appreciate it if you could reconsider your evaluation should our responses address your concerns.

---

> ### Author Response · Authors · 2025-11-28
>
> Dear Reviewer k9nm,
>
> We hope that our responses have adequately addressed your concerns. As the deadline for this discussion phase is approaching, we warmly welcome any further discussion or clarifications regarding additional concerns you may have. We have further clarified our main contributions in previous messages, and Reviewer Uvpd also commented that “the paper's core data efficiency claim (30k vs 10M+) looks solid now.” Could you please confirm whether our rebuttal has addressed your concerns?
>
> Best,
>
> Authors

---

### Author Response · Authors · 2025-11-25
**Looking forward to discussion**

Dear Reviewers,

I hope this message finds you well. It has been about a week since we submitted our rebuttal, and we wanted to kindly follow up to confirm whether our responses have satisfactorily addressed your concerns.

If there are any remaining issues or additional feedback you would like us to consider, please feel free to let us know. Your insights are extremely valuable to us, and we are committed to further improving our work.

Thank you very much for your time and effort in reviewing our paper.

Best regards,

The Authors

---

### Author Response · Authors · 2025-11-27
**Friendly Reminder: Discussion for Paper 3777**

Dear Reviewers,

I hope this message finds you well. **It has been about 2 weeks** since we submitted our rebuttal, and we wanted to kindly follow up to confirm whether our responses have satisfactorily addressed your concerns.

If there are any remaining issues or additional feedback you would like us to consider, please feel free to let us know. Your insights are extremely valuable to us, and we are committed to further improving our work.

Thank you very much for your time and effort in reviewing our paper.

Best regards,

The Authors

---

### Author Response · Authors · 2025-12-01
**Summary for AC: Rebuttal and Reviewer Engagement**

Dear AC,

Thank you very much for your and the reviewers’ time in evaluating our work, and we sincerely hope you could take this into account when forming an overall decision, as **3 reviewers exhibit significant factual misunderstandings of our work and have not engaged with our rebuttal despite multiple polite reminders during the discussion period (from 11.15 to 11.27)**.

Reviewer Uvpd’s main concern was our reported average response length (~5000 tokens). We clarified that this number was a typo and actually refers to character count rather than token count; we have corrected this in the revised version. The true average length is about 1,300 tokens, which is close to realistic conversation length and safely within the reward model’s context window limit (<32,000 tokens). After this clarification, the reviewer confirmed that their main objection was resolved and raised the score to 6 in 11.15.

Reviewer k9nm’s **only stated weakness is about the contribution, with an overall score of 2**. In our work, our main contribution is to propose a data- and compute-efficient synthetic dataset, PiKa, which enables post-training from base models “from scratch” with **10× less SFT data than SoTa instruction datasets**, while achieving greater improvements on widely used alignment benchmarks (AlpacaEval 2.0, Arena-Hard) across multiple backbones such as Llama-3-8B-Base and Qwen-2.5-0.5B/1.5B/7B-Base. To our knowledge, **PiKa is also the first SFT-only dataset that allows these base models to outperform their official instruction-tuned counterparts (eg, Llama-3-8B-Instruct, which is trained on 10M+ data)**, and it shows promising gains on five real-world tasks such as MMLU, TruthfulQA, and others. In addition, PiKa explicitly optimizes prompt diversity, difficulty, and prompt–response quality so that each example is high-signal and non-trivial, which we believe constitutes a substantial and practically important dataset contribution.

Reviewer qzhQ **had a significant factual misunderstanding** of our work’s performance. He cited a recent synthetic‑data work (MoAA) that fine‑tunes Llama‑3.1‑8B‑Instruct (already instruction‑tuned on large proprietary data), while our default setting aligns Llama‑3‑8B‑Base with no prior instruction tuning. This **mixes fundamentally different initialization regimes**. In our rebuttal, we (1) applied PiKa to Llama‑3.1‑8B‑Instruct and showed that PiKa achieves higher evaluation scores than the cited method (2) retrained Llama‑3‑8B‑Base using the MoAA SFT data under our setup and showed that PiKa remains stronger in performance along with 2 times less dataset size;  (3) to additionally show the effect of difficulty and persona, we performed controlled ablations with three different difficulty levels and with/without persona‑based generation settings, further confirming that higher prompt difficulty and persona conditioning consistently improve performance. These results directly address the reviewer’s concerns about both performance and scalability.

Reviewer Et4v also **has a significant factual misunderstanding** regarding the domain and evaluation setting. He questioned the use of a single strong reward model and the reliance on AlpacaEval/Arena‑Hard, and suggested missing “real‑world” tasks and length bias. In our rebuttal, we clarified that using a single high‑quality reward model to construct DPO/RLHF pairs is **standard practice in recent top ML work**, and our reward model is top‑ranked on RewardBench and trained on large‑scale human preferences. We pointed out that our **original paper already reports a broad suite of real‑world benchmarks** (MMLU, ARC‑Challenge, HellaSwag, TruthfulQA, WinoGrande, GSM8K), showing that PiKa‑aligned models maintain or improve general capabilities beyond AlpacaEval and Arena‑Hard. To address LLM‑judge bias, we added multi‑judge evaluation with GPT‑5 and GPT‑4.1‑mini and observed consistent improvements of PiKa over both base and strong baselines. Finally, our analysis shows that PiKa‑aligned models actually produce **slightly shorter responses than some baselines while achieving higher length‑controlled win rates**, indicating that the gains come from better alignment and quality rather than verbosity.

We respectfully feel that our rebuttal has directly addressed the main factual concerns and clarified the key contributions of PiKa as a data‑efficient, transparent benchmark for post‑training from base models. We would be very grateful if you could consider these clarifications, together with Reviewer Uvpd’s updated positive assessment, when making the final decision.

Best,

The Authors

---

### Meta-Review · Area_Chair_ZTH1 · 2025-12-18

**Summary:**

The authors of this submission developed an automated pipeline for synthesizing post-training alignment data. The synthesis is realized via querying powerful LLMs for both prompts and responses, GPT-4o in this work, and a predefined reward model. Experiment results demonstrated the effectiveness of the synthesized data in alignment.

The main concerns of the reviewers mostly felt onto the novelty and technical contribution of this work. As most reviewers pointed out, the solution framework is not conceptually new: prompting a powerful LLM to generate queries, via preset persona or domain knowledge, and then pairing the responses for DPO training, or choosing the top scored one for SFT training. As the authors acknowledged, this distillation based synthetic data generation has been studied for quite a while, but the intellectual improvement against existing solutions is marginal.

As a conclusion, we do not recommend the work for acceptance.

**Reviewer Concerns:**

The typo about the generation length and the starting point of post-training should be addressed; however, the concerns on technical contribution and significance still remain.

**Reviewer Scores:**

I would not believe so.

---

### Decision · Program_Chairs · 2026-01-26

Reject